# Dog Owners’ Knowledge about Rabies and Other Factors That Influence Canine Anti-Rabies Vaccination in the Upper East Region of Ghana

**DOI:** 10.3390/tropicalmed4030115

**Published:** 2019-08-18

**Authors:** Baba Awuni, Elvis Tarkang, Emmanuel Manu, Hubert Amu, Martin Amogre Ayanore, Fortress Yayra Aku, Sorengmen Amos Ziema, Samuel Adolf Bosoka, Martin Adjuik, Margaret Kweku

**Affiliations:** 1School of Public Health, University of Health and Allied Sciences, Ho PMB 31, Volta Region, Ghana; 2Department of Population and Behavioural Sciences, School of Public Health, University of Health and Allied Sciences, Ho PMB 31, Volta Region, Ghana; 3Department of Family and Community Health, University of Health and Allied Sciences, Ho PMB 31, Volta Region, Ghana; 4Department of Epidemiology and Biostatistics, University of Health and Allied Sciences, Ho PMB 31, Volta Region, Ghana

**Keywords:** dog bite, dog owners, rabies, knowledge, vaccination, upper east region, Ghana

## Abstract

Background: Human rabies, often contracted through dog bites, is a serious but neglected public health problem in the tropics, including Ghana. Due to its high fatality rate, adequate knowledge and vaccination of domestic dogs against the disease are very crucial in reducing its burden. We examined dog owners’ knowledge level on rabies and factors that influenced anti-rabies vaccination of dogs in the Upper East Region of Ghana. Methods: This descriptive cross-sectional study was conducted among 260 randomly sampled dog owners in six communities from six Districts using a multistage sampling technique, in the Upper East Region of Ghana. An interviewer-administered questionnaire was used to collect data from the respondents. Descriptive and inferential analyses were done using STATA 14.1. Results: While knowledge about rabies was 199 (76.5%), that about anti-rabies vaccination was 137 (52.7%). District of residence (χ^2^ = 112.59, *p* < 0.001), sex (χ^2^ = 6.14, *p* = 0.013), education (χ^2^ = 20.45, *p* < 0.001) as well as occupation (χ^2^ = 11.97, *p* = 0.007) were significantly associated with rabies knowledge. District of residence (χ^2^ = 57.61, *p* < 0.001), Educational level (χ^2^ = 15.37, *p* = 0.004), occupation (χ^2^ = 11.66, *p* = 0.009), religion (χ^2^ = 8.25, *p* = 0.016) and knowledge on rabies (χ^2^ = 42.13, *p* < 0.001) were also statistically associated with dog vaccination against rabies. Dog owners with good knowledge on rabies for instance, were more likely to vaccinate their dogs against rabies compared to those with poor knowledge [AOR = 1.99 (95% CI: 0.68, 5.86), *p* = 0.210]. Dog owners with tertiary level of education were also 76.31 times more likely (95% CI: 6.20, 938.49, *p* = 0.001) to have good knowledge about rabies compared to those with no formal education. Conclusions: Dog owners in the Upper East Region of Ghana had good knowledge about rabies. This, however, did not translate into correspondingly high levels of dog vaccination against the disease. Rabies awareness and vaccination campaigns should, therefore, be intensified in the region, especially among the least educated and female dog owners.

## 1. Introduction

Dog bites are considered a public health concern worldwide as they cause physical injury, psychological trauma, post-traumatic stress, and sometimes death resulting from rabies infection [1,2]. Rabies, a neglected tropical disease, is a neuro-invasive disease caused by Rhabdovirus, commonly transmitted from the saliva of some infected warm-blooded animals such as domestic dogs and cats, through bites or other forms of contact [3,4]. About 99% of human rabies cases are transmitted by domestic dogs [5]. Though rabies is a vaccine-preventable viral zoonosis, it remains a critical public health issue, with about 59,000 human deaths reported globally each year, with Asia and Africa accounting for more than 95% of these deaths [6,7].

Ghana is one of the countries on the African continent where rabies remains endemic [8]. Rabies is a public health concern in Ghana due to a large stray dog population across the country [9]. Despite efforts to reduce the prevalence of rabies in the country, substantial numbers of new cases are recorded annually. From the year 2000 to 2004, a total of 123 clinically confirmed rabies cases were reported [10]. Between January 2009 and July 2011, 25 rabies cases resulting from canine bites were recorded in the country, marking a significant reduction in the number of rabies deaths in the country [10]. However, as a result of erratic rabies prevention strategies; thus inconsistent rabies prevention measures, rabies cases peaked up again by the year 2016, where 64 human rabies cases were recorded [10]. This is because although the Ghana government embarked on a campaign to vaccinate dogs against rabies across the country [11] the campaign was not sustained and has been stopped since 1994 due to lack of sustainable funding and support. Since then, individual dog owners are expected to use their personal income to vaccinate their dogs against rabies either by the government or private veterinary officers [12]. Although there is no documented standardized pricing for anti-rabies vaccination of dogs in Ghana, information from the veterinary division of the Ministry of Agriculture revealed that a shot of anti-rabies vaccine can cost up to $4 per dog. This places huge financial burden on dog owners, especially in a country where 23.4% of the population live on less than $1 a day [13]. Hence, increased financial burden, coupled with the sporadic nature of anti-rabies vaccine and post exposure prophylaxis supply [12,14] has exacerbated the risk of potential rabies outbreak, therefore, the need for dog owners to be knowledgeable about rabies and the need to vaccinate dogs against rabies.

The Upper East Region is the only known region out of the 10 regions in the country where dogs are sold in open markets for human consumption [15], in addition to regular dog keeping. However, these dogs are not properly confined, which results in persistent straying of dogs [16]. There is therefore, the likelihood of frequent dog bites, which could lead to the possibility of high number of rabies cases as the percentage of dogs vaccinated against the disease remains unknown [17]. Rabies affects the central nervous system, with almost 100% case fatality rate [8]. As such, knowledge of dog owners on rabies, anti-rabies vaccination coverage, and factors that underline anti-rabies vaccination coverage in a region where dogs are sold on the open market for consumption, are not confined and also rely on individual dog owners to fund anti-rabies dog vaccination, could be useful prerequisites in developing effective rabies control strategies in the Upper East Region and Ghana as a whole. The current study, therefore, ascertained dog owners’ knowledge on rabies and factors influencing anti-rabies vaccination by dog owners in the Upper East Region of Ghana.

## 2. Methods

### 2.1. Setting

Ghana is a country in the Western part of Africa with 10 administrative regions; Greater Accra, Western, Eastern, Central, Volta, Ashanti, Brong Ahafo, Northern, Upper West, and Upper East regions. The Upper East Region is located in the North-Eastern corner of the country and between longitudes 0° and 1° West and latitudes 100°30″N and 110N with Bolgatanga as its capital. It has two international boundaries; Burkina Faso to the North and the Republic of Togo to the East; and also shares borders with the Northern and Upper West Regions of Ghana [18]. The 2010 Population and Housing Census of Ghana puts the population of the Region at 1,046,545 [18]. Figure 1 presents a map of the study setting.

### 2.2. Study Design

This descriptive cross-sectional study was conducted in April 2017 among dog owners in six communities from six districts in the Upper East Region of Ghana. The cross-sectional nature of the study aided to obtain the needed information from the respondents at a single point in time although recruitment of respondents took place over a period of time [19].

#### Study Population and Sampling

The source population comprised all adult (18 years and above) dog owners within the Upper East Region of Ghana, who had resided in the region for a minimum of 6 months. A representative sample of 260 respondents from six selected districts with an anticipated attrition (turnover) rate of 5%, based on experiences drawn from the pilot study [20], was used for the study. This was calculated using the formula *n* = $Z^{2}$ × *P* (1 − *P*)/$E^{2}$ [21].

A three-stage sampling technique was used for the selection of the respondents from six out of the 15 districts in the region. The first stage consisted of selection of six Districts from the fifteen districts in the region. This was done by randomly selecting from the list of districts, using the simple random (lottery) sampling technique. With the lottery approach, the names of the districts were written on pieces of paper, put in a container, thoroughly shaken and the first district selected. The process was then repeated until the 6th district was selected. The second stage was the selection of communities per district which made use of the same method adopted in selecting the districts. The selected communities were Sumburungu in Bolgatanga Municipality, Binaba in Bawku West District, Gore in Bongo District, Abiliyire No. 1 in Builsa North District, Wiidi in Bawku Municipality and Natugnia in Kassena Nankana Municipality.

The final stage involved the selection of households and dog owners. Since a list of households and dog owners were not available, households were selected per community based on the WHO cluster survey technique [22]. A central location in each town was first identified. A starting point was then randomly selected from the centre using the bottle method. With this, a bottle was spun and the direction where the mouth of the bottle settled was selected. The research team then walked in the selected direction, counting the number of houses until the edge of the village was reached. Using a systematic sampling approach, the sampling fraction of 5 was selected from a list of total number of houses counted. Each 5th house from the centre along the chosen direction was then chosen. In households where there were more than one dog owners, and were both willing to participate, one dog owner was selected by ballot. Households with no dog owners were skipped. The next household with a dog owner was chosen to replace the skipped household. A pre-tested questionnaire was used by trained research assistants for data collection.

### 2.3. Data Analysis

Data collected were analysed using STATA 14.1 (StataCorp LLC, College Station, TX, USA). Respondents’ knowledge on rabies was assessed by asking a set of six questions on rabies relating to its causes, symptoms and modes of transmission, adapted from Assefa et al. [23]. The questions asked were: Do dogs carry rabies virus? Can a dog transmit rabies to humans through biting or scratching? Can a dog transmit rabies to humans through licking of open wound? Are aggression, madness and paralysis all signs of rabies infection in humans? Can rabies be prevented? Is there the need to go to the hospital after being bitten by a dog? The questions were chosen based on their ease of understanding to respondents during piloting as well as their acceptable reliability (Cronbach’s alpha of 0.94) in measuring respondents’ level of knowledge on rabies, as per Taber’s interpretation [24].

In determining whether respondents had good or poor knowledge on rabies, the median score of the six questions, 4, was used. A score equivalent to the median or above was rated as good knowledge while below that was considered as poor knowledge [25]. Chi-square tests were conducted to determine the relationship between socio-demographic characteristics and knowledge on rabies. Binary logistic regression models were then used to determine the strength of association between the socio-demographic characteristics and knowledge on rabies and vaccination of dogs. All statistical analyses were considered significant at *p*-value < 0.05. The results were then presented in tables and graphs.

### 2.4. Ethical Issues

Ethical approval was obtained from the University of Health and Allied Sciences (UHAS) Research Ethics Committee (UHAS-REC) (UHAS-REC A.6 [14] 17–18). Permission was also sought from the Upper East Regional Health Directorate and the municipal/district health directorates where data were collected. A written informed consent was obtained from the participants and confidentiality and anonymity were assured.

## 3. Results

### 3.1. Socio-Demographic Characteristics of Respondents

Out of the total number of 260 respondents, 186 (71.5%) of them were males and 122 (46.9%) were without any form of formal education. Most 190 (73.1%) were married, with farming being the predominant occupation (158, 60.8%) amongst them (Table 1).

### 3.2. Knowledge of Dog Owners on Rabies

Table 2 presents the responses to various variables on awareness, transmission and treatment of rabies that were used to determine good and poor knowledge on rabies. Majority 214 (82.3%) knew that dogs were susceptible hosts of rabies, with 209 (80.4%) also agreeing that dogs transmit rabies through bites and scratching. However, few 9 (3.5%) knew that aggression, madness and paralysis were all signs of rabies. Majority (195, 75.0%) and (7204, 8.5%) also knew that rabies is preventable and that one has to go to the hospital for treatment after being bitten by a rabid dog respectively.

Overall, we realized that 199 (76.5%) of the dog owners had good knowledge on rabies, based on the criteria used.

### 3.3. Association between Socio-Demographic Characteristics and Knowledge Level of Dog Owners on Rabies

From Table 3, significant associations were found between dog owners’ knowledge about rabies and district of residence (χ^2^ = 112.59, *p* < 0.001), sex (χ^2^ = 6.14, *p* = 0.013), education (χ^2^ = 20.45, *p* < 0.001) as well as occupation (χ^2^ = 11.97, *p* = 0.007). Dog owners in the Bongo District (AOR = 6.74, (95% CI: 1.47, 30.89), *p* = 0.014) and the Bolga Municipality (AOR = 7.89, (95% CI: 1.49, 41.71) *p* = 0.015) were significantly more likely to have good knowledge compared to those in the Bawku Municipality. On the contrary, dog owners in the Kassena Nankana District were 96% significantly less likely to have good knowledge as compared to those in the Bawku (95% CI: 0.01, 0.17, *p* < 0.001). Dog owners with tertiary level of education were also 76.31 times more likely (95% CI: 6.20, 938.49, *p* = 0.001) to have good knowledge on rabies compared to those with no formal education.

### 3.4. Vaccination of Dogs by Owners against Rabies

It was realized that 137 (52.7%) of dog owners had currently (within a year at the time of data collection) vaccinated their dogs against rabies.

#### 3.4.1. Association between Socio-Demographic Characteristics and the Vaccination of Dogs againts Rabies

Table 4 shows that there was a significant association between district (χ^2^ = 57.61, *p* < 0.001), level of educational attainment (χ^2^ = 15.37, *p* = 0.004), occupation, (χ^2^ = 11.66, *p* = 0.009), religion (χ^2^ = 8.25, *p* = 0.016) and vaccination of dogs. Dog owners with tertiary level of education were more likely to vaccinate their dogs against rabies compared to those with no education (AOR = 3.57, 95% CI: 0.97, 13.14).

#### 3.4.2. Association between Awareness and Knowledge on Rabies and the Vaccination of Dogs against Rabies

There was a significant association between hearing about rabies, having good knowledge aboutrabies and vaccination of dogs against rabies (χ^2^ = 46.25, *p* < 0.001) and (χ^2^ = 42.13, *p* < 0.001) respectively (Table 5). Dog owners who had heard of rabies were 12 times more likely to vaccinate their dogs against the disease compared to those who had never heard of it (95% CI: 2.54, 59.15, *p* = 0.002). Even though not statistically significant, dog owners with good knowledge on rabies were more likely to vaccinate their dogs against rabies compared to those with poor knowledge [AOR = 1.99 (95% CI: 0.68, 5.86), *p* = 0.210].

## 4. Discussion

The findings in this study revealed that the majority of dog owners (76.5%) had good knowledge on rabies, its cause, symptoms, effects and prevention of the disease. Contrary to our findings other studies conducted in Ethiopia [23] and Tanzania [26] found knowledge about rabies to be 49.5% and 37% which were respectively much lower. The high level of knowledge we observed could be attributed to the fact that since the Upper East Region is a region where dogs are openly sold on the market for human consumption [15], a lot of the population was probably familiar with domestic dogs, including diseases like rabies that could be contracted from their handling and consumption.

On whether dogs were vaccinated against rabies, 52.7% of the respondents answered in the affirmative. Thus, the good knowledge about rabies demonstrated by the respondents did not translate into adoption of desirable preventive measures against the disease. Considering the high fatality rate of rabies, the reported percentage of dog vaccination is not encouraging [7,16]. Moreover, the American Animal Hospital Association’s 2017 canine vaccination guidelines mentioned rabies vaccine as a core vaccine for dogs in rabies endemic regions such as Ghana. Thus, it is a requirement for every dog in such regions or countries to be vaccinated against rabies within the first six months of life. They then need a booster one year after the first vaccination and thereafter, should be vaccinated every three years. However, it is recommended that dogs be vaccinated on a yearly basis in regions where rabies is endemic [27]. Hence, the percentage rabies vaccination coverage found in this study, was woefully inadequate if rabies is to be controlled. Although there are no specific guidelines for dog vaccination against rabies in Ghana, according to Lopes et al., about 70% of the dog population in the country need to be vaccinated against rabies before a significant reduction in human rabies cases can be achieved [12]. However, the issue of low immunization of dogs against rabies is not peculiar to Ghana and specifically, the Upper East Region. Though higher than our recorded figure, dog vaccination against the disease has been reported to be below 80% in Brazil [28]. Comparing our finding to that of [28] affirms the low percentage vaccination of dogs in the Upper East Region of Ghana where dogs are not only kept as pets, but are also sold in the open market as meat. Considering the inhumane handling of dogs by both sellers and dog meat processors [29], one could easily be bitten by a dog that senses danger or feels distressed.

Our study further revealed that district of residence of dog owners influenced knowledge on rabies. Dog owners from Bolgatanga Municipality were for instance 7.89 times more likely to have good knowledge about rabies than those from Bawku Municipality. This result could be attributed to finding of the 2010 population and housing census where the Bolgatanga Municipality recorded the highest literacy rate (54.61%) in the region [17]. Thus, the high level of education probably translated into high knowledge on rabies since education has been noted as an important predictor of health literacy in the literature [30,31] as well as our own findings.

The high level of knowledge on rabies in the Bolgatanga Municipality did not translate into the practice of vaccinating dogs against rabies in the municipality. We found that residents of the municipality were 29% less likely to vaccinate their dogs against rabies as compared to the reference group, Bawku Municipality. In trying to understand knowledge, attitude and practice about dog bite and rabies exposure among dog meat consumers and processors in Abia State, Nigeria [29], found that although dog meat eaters and processors were very knowledgeable about rabies (73.7% and 71.6% respectively), they did not consider rabid dogs to be of public health concern. Instead, they associated these dogs with medicinal and spiritual values, where some parts of rabid dogs were deemed capable of healing ailments such as malaria, enhancement of libido and spiritual protection. Thus, although residents of Bolgatanga Municipal were knowledgeable about rabies, they might not have been vaccinating their dogs against the disease as they might not have considered it to be of public health concern or are likely to have their own beliefs and perceptions of rabies and rabid dogs.

Sex was also found to be associated with knowledge on rabies. Male dog owners were for instance 1.5 times more likely to have good knowledge on rabies than females. Our finding is consistent with a study conducted by [32] who found that on the average, few women (4.1%) are knowledgeable about the mode of transmission of some selected zoonotic diseases, including rabies. Thus, although knowledge acquisition is positively skewed in favour of women under equal socio-environmental conditions [33,34,35,36], it is not the case in the Bolga Municipality when it comes to knowledge on zoonotic diseases such as rabies. However, the low knowledge level of women on rabies in our study could be as a result of low interest in dog ownership among Ghanaian women in general [37] and in the Upper East Region in particular, as only 28.5% of our respondents were females. Thus, while American women are more likely to own dogs [38], that might not be the case in Ghana, especially in the Upper East Region. Hence, the less interest of females in the Upper East Region to own dogs might have affected their interest in knowing much about dogs and the diseases they transmit, such as rabies.

Occupation and religious affiliation of dog owners were found to significantly influence decision to vaccinate dogs against rabies. Dog owners who were traders, public and civil servants or farmers were respectively 3.4, 1.3 and 1.1 times more likely to vaccinate their dogs against rabies than those who were unemployed. Also, dog owners who were Muslims were 2.6 times more likely to vaccinate their dogs against rabies as compared to Christian dog owners. Moreover, dog owners who were traditionalists, which is an African way of understanding God and the world by believing in supernatural world of spirits and powers [39], were 4% less likely to vaccinate their dogs against rabies compared to Christians. According to the WHO [40], despite the introduction of Western medicine and healthcare systems in Africa, many Africans still rely on traditional medicine, often practiced by traditionalists. Most of these traditionalists thus do not believe in Western Medicine as they trust their supernatural powers to protect them from disease and illness and often attributes ill-health to a curse or punishment from the supernatural world [41]. Hence, they are likely not to blame rabies on dogs but see it as a misfortune or punishment from the spiritual world. As a result, traditionalists are less likely to vaccinate their dogs against rabies. The association between employment status and dog vaccination against rabies found in our study could be explained by the fact that rabies vaccine is administered at a cost to the dog owner as there are no funds for free vaccination of dogs in Ghana [42]. Hence dog owners who are unemployed may be reluctant to vaccinate their dogs against the disease as they may not be able to afford the cost of the vaccine. This has been explained by [43] that socio-economic conditions of individuals influence their health seeking decisions and behaviour.

The higher likelihood of Muslims vaccinating their dogs against Christian dog owners could be as a result of their faith. Islam does not require treatment to be provided to a Muslim patient if it merely prolongs the span of a terminal illness [44]. Buttressing this assertion, [45] explained that Islam attached significant importance to health; as such taking care of oneself is a religious duty. Hence, with such belief systems and the fatality of rabies in mind, Muslim dog owners might find it appropriate to vaccinate their dogs so as to prevent the case of managing a fatal illness like rabies by vaccinating their dogs against it to protect their much-cherished good health as a religious obligation.

Our study found that educational level significantly influenced knowledge of dog owners on rabies. Dog owners who had tertiary education had higher odds of having good knowledge on rabies and vaccinating their dogs against it than those with no formal education. Hence, high level of education of dog owners translated into good vaccination practices of dogs against rabies. Moreover, we found that dog owners who ever heard of rabies were 12.3 times more likely to vaccinate their dogs against the disease than those who did not. This is not surprising as education has been noted as an important predictor of health decision making [46]. This finding is consistent with studies done in Tanzania and Pakistan where respondents with secondary and tertiary education were associated with having better knowledge of rabies [47,48]. Therefore, awareness creation and education on rabies among dog owners is should be encouraged among dog owners in the Upper East Region of Ghana to improve vaccination of dogs against the diseases. This could be achieved by training and awarding of certificates to individuals such as unemployed graduates, to becoming active members by educating communities about rabies, as is the case in South Africa, Lesotho Zimbabwe and Mozambique [49]. Furthermore, village chiefs, community leaders, traditional healers and teachers could be engaged in rabies awareness and information counseling as they are more trusted in their communities than health workers and veterinary officers who are most often not indigenous community members. Also, awareness campaigns could target school children through development, dissemination and usage of children’s rabies awareness booklets to be used by teachers in schools as part of the school curriculum to educate children on the prevalence, symptoms effects and prevention methods on rabies in schools, as in the case of Zimbabwe [49].

## 5. Limitations of the Study

Despite the important findings we made in this study, the limitations of the study are worth noting. One of the limitations with this study is the non-inspection of vaccination certificates of the dog owners. We believe that this could have corroborated the self-reports given by the participants regarding their vaccination of the dogs and added more value to the study.

Also, since our study was cross sectional in nature, we did not establish unobserved heterogeneity. Thus, a causal relationship could not be established between dog vaccination and knowledge as well as factors identified other than, an association. Also, the associations observed between the outcome and explanatory variables could also vary over time. Moreover, the high level of knowledge we recorded could be as a result of the wording of some of the questions which potentially led respondents to the correct answers. Thus, the knowledge level could have been lower if more neutral questions were asked. Lastly, we did not collect data on respondents’ income status, hence could not fully explore the role socio-economic status could play in influencing decisions on dogs’ vaccination against rabies.

## 6. Conclusions

Level of knowledge about rabies was found to be associated with the district of residence, sex, educational level as well as occupation of dog owners. District of residence, educational level, occupation, religion and knowledge on rabies also had significant association with dog vaccination against rabies. Given that dog owners in the Upper East Region of Ghana had good knowledge about rabies, but this did not translate into correspondingly high levels of dog vaccination against the disease, we recommend that rabies awareness and vaccination campaigns should be intensified in the region, especially among the least educated and female dog owners. Also, future studies should take into consideration the role of socio-economic status in influencing anti-rabies dog vaccination decisions in the region.

## Figures and Tables

**Figure 1 tropicalmed-04-00115-f001:**
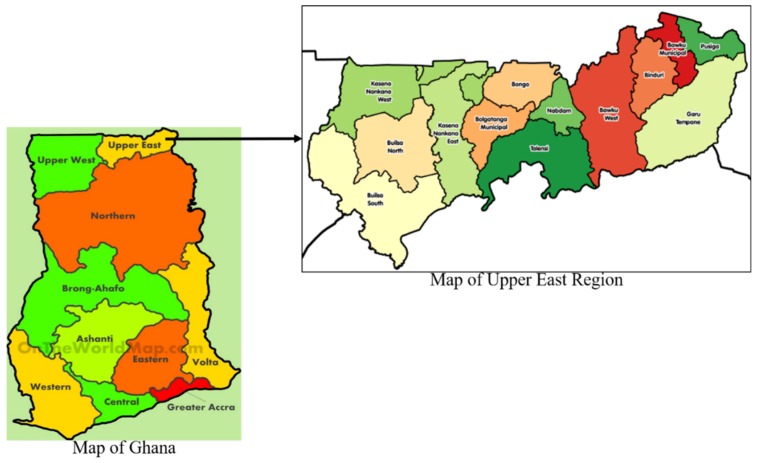
Study setting.

**Table 1 tropicalmed-04-00115-t001:** Socio-demographic characteristics of respondents.

Variable	Frequency [*n* = 260]	Percent (%)
Age (In completed years)		
<20	9	3.4
20–29	66	25.4
30–39	53	20.4
40–49	66	25.4
50+	66	25.4
Sex		
Male	186	71.5
Female	74	28.5
Education		
None	122	46.9
Primary	35	13.5
JHS	30	11.5
SHS	34	13.1
Tertiary	39	15.0
Occupation		
Unemployed	31	11.9
Trader	37	14.2
Farmer	158	60.8
Civil/public servant	34	13.1
Marital status		
Never married	51	19.6
Married	190	73.1
Widowed	19	7.3

**Table 2 tropicalmed-04-00115-t002:** Knowledge of dog owners about rabies.

Variable	Frequency (*n*)	Percentage (%)
Do dogs carry rabies virus? (*n* = 260)		
Yes	214	82.3
No	46	17.7
Can a dog transmit rabies to human through biting or scratching? (*n* = 214)		
Yes	208	97.2
No	6	2.8
Can a dog transmit rabies to human through licking of open wounds? (*n* = 214)		
Yes	33	15.4
No	181	84.6
Are aggression, madness and paralysis all signs of rabies infection in humans? (*n* = 260)		
Yes	9	3.5
No	251	96.5
Can rabies be prevented? (*n* = 260)		
Yes	195	75.0
No	65	25.0
Is there the need to go to the hospital after being bitten by a dog? (*n* = 260)		
Yes	204	78.5
No	56	21.5

**Table 3 tropicalmed-04-00115-t003:** Association between socio-demographic characteristics and knowledge about rabies.

Variable	Poor Knowledge [*n* = 61] *n* (%)	Good Knowledge [*n* = 199] *n* (%)	Total [*n*= 260] *n* (%)	χ^2^ (*p*-Value)	COR (95% CI) *p*-Value	AOR (95% CI) *p*-Value
District						
Bawku Municipality	9 (14.7)	37 (18.6)	46 (17.7)	112.59 (<0.001)	Ref.	Ref.
Bawku West District	4 (6.6)	39 (19.6)	43 (16.5)		2.37 (0.67, 8.37) 0.179	3.78 (0.91, 15.77) 0.068
Bolga Municipality	2 (3.3)	42 (21.1)	44 (16.9)		5.11 (1.04, 25.17) 0.045	7.89 (1.49, 41.71) 0.015
Bongo District	3 (4.9)	44 (22.1)	47 (18.1)		3.57 (0.90, 14.15) 0.070	6.74 (1.47, 30.89) 0.014
Builsa North District	6 (9.8)	30 (15.1)	36 (13.9)		1.22 (0.39, 3.80) 0.736	0.41 (0.09, 1.87) 0.249
Kassena Nankana District	37 (60.7)	7 (3.5)	44 (16.9)		0.05 (0.02, 0.14) 0.000	0.04 (0.01, 0.17) <0.001
Sex						
Female	25 (41.0)	49 (24.6)	186 (71.5)	6.14 (0.013)	Ref.	Ref.
Male	36 (59.0)	150 (75.4)	74 (28.5)		2.13 (1.16, 3.89) 0.014	1.54 (0.60, 3.96) 0.374
Education						
None	42 (68.9)	80 (40.2)	122 (46.9)	20.45 (<0.001)	Ref.	Ref.
Primary	8 (13.1)	27 (13.6)	35 (13.5)		1.77 (0.74, 4.24) 0.199	5.80 (1.46, 23.07) 0.013
JHS	6 (9.8)	24 (12.0)	30 (11.5)		2.10 (0.80, 5.54) 0.134	2.21 (0.52, 9.33) 0.281
SHS	4 (6.6)	30 (15.1)	34 (13.1)		3.94 (1.30, 11.93) 0.015	8.80 (1.82, 42.50) 0.007
Tertiary	1 (1.6)	38 (19.1)	39 (15.0)		19.95 (2.65, 150.45) 0.004	76.31 (6.20, 938.49) 0.001
Occupation						
Unemployed	3 (4.9)	28 (14.0)	31 (11.9)	11.97 (0.007)	Ref.	
Farmer	48 (78.7)	110 (55.3)	37 (14.2)		0.25 (0.07, 0.85) 0.026	0.54 (0.10, 2.83) 0.468
Trader	7 (11.5)	30 (15.1)	158 (60.8)		0.46 (0.11, 1.95) 0.292	0.49 (0.06, 3.73) 0.487
Civil/public servant	3 (4.9)	31 (15.6)	34 (13.1)		1.11 (0.21, 5.94) 0.905	0.16 (0.02, 1.37) 0.095

**Table 4 tropicalmed-04-00115-t004:** Association between socio-demographic characteristics and the vaccination of dogs against rabies and the odds of dog vaccination against rabies.

Variable	Vaccinated Dog		Total [*n* = 260] *n* (%)	χ^2^ (*p*-Value)	COR (95% CI) *p*-Value	AOR (95% CI) *p*-Value
	No [*n* = 123] *n* (%)	Yes [*n* = 137] *n* (%)				
District						
Bawku Municipal	18 (14.6)	28 (20.4)	46 (17.7)	57.61 (<0.001) **	Ref.	
Bawku West	11 (8.9)	32 (23.4)	43 (16.5)		1.87 (0.76, 4.63) 0.175	5.27 (1.73, 16.03) 0.003
Bolga Municipal	24 (19.5)	20 (14.6)	44 (16.9)		0.54 (0.23, 1.24) 0.144	0.79 (0.28, 2.22) 0.660
Bongo	17 (13.8)	30 (21.9)	47 (18.1)		1.13 (0.49, 2.63) 0.768	2.92 (1.02, 8.35) 0.045
Builsa North	11 (8.9)	25 (18.2)	36 (13.9)		1.46 (0.58, 3.68) 0.421	3.31 (1.02, 10.71) 0.046
Kassena Nankana	42 (34.2)	2 (1.5)	44 (16.9)		0.03 (0.01, 0.14) 0.000	0.06 (0.11, 0.29) 0.001
Education						
None	68 (55.3)	54 (39.4)	122 (46.9)	15.37 (0.004) **	Ref.	Ref.
Primary	16 (13.0)	19 (13.9)	35 (13.5)		1.50 (0.70, 3.18) 0.296	2.16 (0.83, 5.62) 0.114
JHS	13 (10.6)	17 (12.4)	30 (11.5)		1.65 (0.74, 3.69) 0.225	1.27 (0.47, 3.45) 0.635
SHS	18 (14.6)	16 (11.7)	34 (13.1)		1.12 (0.52, 2.40) 0.772	0.62 (0.23, 1.65) 0.334
Tertiary	8 (6.5)	31 (22.6)	39 (15.0)		4.88 (2.07, 11.48) 0.000	3.57 (0.97, 13.14) 0.056
Occupation						
Unemployed	14 (11.4)	17 (12.4)	31 (11.9)	11.66 (0.009) **	Ref.	Ref.
Farmer	86 (69.9)	72 (52.5)	158 (60.8)		0.69 (0.32, 1.49) 0.346	1.07 (0.41, 2.77) 0.891
Trader	15 (12.2)	22 (16.1)	37 (14.2)		1.21 (0.46, 3.17) 0.701	3.37 (0.94, 12.06) 0.062
Civil/public servant	8 (6.5)	26 (19.0)	34 (13.1)		2.68 (0.93, 7.74) 0.069	1.25 (0.32, 4.87) 0.752
Religion						
Christian	62 (50.4)	80 (58.4)	142 (54.6)	8.25 (0.016) *	Ref.	
Muslim	9 (7.3)	20 (14.6)	29 (11.2)		1.72 (0.73, 4.04) 0.212	2.64 (0.90, 7.73) 0.076
African traditional religion	52 (42.3)	37 (27.0)	89 (34.2)		0.55 (0.32, 0.94) 0.030	0.96 (0.46, 2.02) 0.921

Note = * *p* < 0.05, ** *p* < 0.01.

**Table 5 tropicalmed-04-00115-t005:** Association between awareness and knowledge on rabies and the vaccination of dogs against rabies and the odds of dog vaccination.

Variables	Vaccinated Dog		Total [*n* = 260] n (%)	χ^2^ (*p*-Value)	COR (95% CI) *p*-Value	AOR (95% CI) *p*-Value
	No [*n* = 123] n (%)	Yes [*n* = 137] n (%)				
Ever heard of rabies						
No	42 (34.1)	3 (2.2)	45 (17.3)	46.25 (<0.001)	Ref.	Ref.
Yes	81 (65.9)	134 (97.8)	215 (82.7)		23.16 (6.95, 77.15) <0.001	12.25 (2.54, 59.15) 0.002
Knowledge on rabies						
Poor	51 (41.5)	10 (7.3)	61 (23.5)	42.13 (<0.001)	Ref.	Ref.
Good	72 (58.5)	127 (92.7)	199 (76.5)		9.0 (4.31, 18.80) <0.001	1.99 (0.68, 5.86) 0.210

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
