# Peer review of "Dog Owners’ Knowledge about Rabies and Other Factors That Influence Canine Anti-Rabies Vaccination in the Upper East Region of Ghana"

_tropicalmed, 2019, doi:10.3390/tropicalmed4030115_

Round 1

Reviewer 1 Report

This article is interesting and well written. It gives some information about the level of knowledge of the dog owner regarding the rabies disease. This kind of study could be enlarged to other regions of Ghana or to other countries where rabies is endemic. It could allow to identify if some needs are required for improving the education of people on this topic and some awareness campaigns could be done. This would act as an important prevention mean against rabies for people, as strongly recommended by the International Organizations (OIE, WHO, FAO, GARC).

In this article, the authors have assessed the level on rabies knowlegde of the dog owners in the Upper Est region of Ghana. They have also tried to find if the anti-rabies vaccination of dogs was influenced by some socio-demographic factors. To do that, they have conducted a descriptive cross-sectional study in six communities from six districts randomly chosen. To conclude, the authors have emphazised the fact that rabies awareness and vaccination campaigns should be improved in the testing area especially for the least educated and female dog owners.

2. Methods

Line 82 : What is the meaning of the word « us » ?

Line 87 : How do the authors fix the turnover rate to 5 % ? Is it based on data collected in the field or chosen arbitrarily by the authors ?

For the data analysis section, it should be good for the reader to know the 6 questions asked to people. Moreover, it could help people to conduct such kind of study in other rabies endemic areas or countries.

3. Results

Line 150 : It could be interesting to know the context of anti-rabies vaccination of dogs in this region (private veterinary or mass vaccination campaign… and the cost per vaccination). This could be added in the introduction.

4. Discussion

Maybe the authors could add some words on how to improve the awereness on rabies and vaccination in the testing area. They could also discuss on what it is already carried out and the obtained results in other countries according to the recommendations of the International Organizations.

Author Response

Hello, 

Please see the attached file for my response to the review comments. 

Thank you very much for your input. 

It is much appreciated.

Regards

Reviewer 2 Report

p.p1 {margin: 0.0px 0.0px 0.0px 0.0px; font: 12.0px Helvetica; color: #000000; -webkit-text-stroke: #000000} p.p2 {margin: 0.0px 0.0px 0.0px 0.0px; font: 12.0px Helvetica; color: #000000; -webkit-text-stroke: #000000; background-color: #ffffff} span.s1 {font-kerning: none; background-color: #f6f6f6} span.s2 {font-kerning: none} span.s3 {font-kerning: none; background-color: #ffffff}

In the abstract, I don’t think the authors can conclude that educational level, occupation, religion, and knowledge on rabies informed the decision to vaccinate. Based on the study design and the analysis, the can merely conclude that they are statistically associated.

In the introduction, the authors have to provide information on the availability of dog rabies vaccine in the area. Are there many animal clinics? Are there mass dog vaccination campaigns? Are dog vaccines free? It is fair to assume people with higher educational level have higher SES. If the vaccines are not free, the association between higher education and higher vaccination might be due to the affordability of the vaccination and not education per se. In the same way, usually, women have a lower income than men. If the vaccines are costly that might explain why female dog owner vaccinate less their animals. In those case, increasing education is not going to change much, because the barrier to vaccinating dog is economic.

Line 47,48. Reference 6 is not the appropriate reference. Use “Hampson, 2015. Estimating the global burden of endemic canine rabies.” Also, the burden of disease is 59,000 deaths, not 60,000.

Line 55. Explain what you mean by ‘erratic rabies prevention strategies’.

Line 60,61. Sentence about encephalitis is out of place in this paragraph.

Lines 55-66. That paragraph is centered about dogs for human consumption and how important this practice is in the upper east region of Ghana. However, at the end of the paragraph, the authors say the objective is to dog owner’s knowledge. Are dog owners' breeding dogs for meat trading? If the purpose of the study is to evaluate regular dog owners (not people breeding dogs for meat consumption), it’d be better if they use the last paragraph to present the importance of their cross-sectional study. 

Line 81. Remove capital letter from ‘Cross’

Line 85. All the adult dog owners are your source population, not your study population. Your study populations are those that you invite to participate in the survey.

Lines 92,93. The authors have access to computers and software program. I suggest next time, use code to randomize and sample your districts and houses so your process can be replicated.

Line 108. When a household with no dogs was skipped, did the authors choose a new house to replace the skipped house?

Lines 112, 113. What do the authors mean when they say they used Assefa et al’s method? Assefa et al. used many more questions than six and they did not propose a new method. 

Lines 113. How did the authors choose the set of six questions for the knowledge questionnaire? how did they determine that 3 out of 6 was the cut off for good knowledge and why are they dichotomizing knowledge that at least have six levels based on their questionnaire? Provide the questions you asked to assess knowledge and the proportion of answers for each question.

Lines 114. The authors have more than 200 surveyed dog owners with multiple variables that are related to knowledge and vaccinating their dogs. Have the authors attempted to build multivariate regression models? I think it’d be important to see if these bivariate associations change when adjusted for other variables.

Line 150. 53% of dogs were vaccinated during the last year? within their lifetime? Also, did the authors captured any data on dog’s age/sex/restriction? did they find any association between those variables and vaccination?

Table 3. Explain what ‘traditional’ means in the Religious categories and also discuss that finding. Seems that most of the traditional dog owners do not vaccinate their dogs.

The authors found important differences in variables that might be related to socioeconomic status (sex, occupation, education, district). It might be interesting if they explore that in their analysis (they can compare income by district and vaccination level by district for example), and if they add to the discussion the potential role of SES in the behavior of vaccinating dogs.

Author Response

Hello

Please see the attached file for my response to the comments raised by the reviewer.

I appreciate your input.

Thank you very much.

Regards

Round 2

Reviewer 2 Report

The authors have responded well most of my comments and the manuscript has improved.

However, 

They have not explained how they chose six questions out of the many questions used by Asefa et al.

They have not provided the list of six questions they asked. In table 2, they present the variables they measured, but not how the question was asked. Please provide the exact questions asked during the survey.

Author Response

Hello

Please see the attached file for my response to the reviewer's comments.

Thank you

Round 3

Reviewer 2 Report

Thank you for providing the exact questions in table 2. Some of the questions used in the questionnaire are 'leading question'. That means the way they are worded could lead the respondents to one side of the argument.

For example, if you want to assess if the respondents know that dogs can carry rabies, you could ask "Which animal/species can carry the rabies virus?". That's a more neutral question compared to the one you asked (Do dogs carry rabies virus?)

After that, if you want to assess if the respondents know that rabid dogs can transmit rabies through biting or scratching, you must ask that question ONLY to those respondents who mentioned dogs in the previous question. However, you included all the respondents in this question, including those who did not mention dogs as carriers of rabies virus.

The second question is also a leading question (Can a dog transmit rabies to human through biting or scratching?). A more neutral question would have been "How can rabid dogs transmit rabies to humans?" That way you are not giving them the answer!

Please, reanalyze your data for questions 2 and 3. If somebody did not mention dogs as carriers/reservoirs/hosts of rabies virus do not include them in the calculations of question 2 and 3.

Also, in limitations, you must disclose that the very high level of knowledge you are reporting could have been lower if more neutral questions were asked.

Author Response

Hello.

Please see the attached file for my response. 

Thank you.

Round 4

Reviewer 2 Report

I appreciate the time the authors invested in responding to my comments and improve the manuscript.